# Low Density Lipoprotein Exposure of Plasmacytoid Dendritic Cells Blunts Toll-like Receptor 7/9 Signaling via NUR77

**DOI:** 10.3390/biomedicines10051152

**Published:** 2022-05-17

**Authors:** Anette Christ, Pieter G. Goossens, Erwin Wijnands, Han Jin, Bart Legein, Tammy Oth, Aaron Isaacs, Monika Stoll, Joris Vanderlocht, Esther Lutgens, Mat J. A. P. Daemen, Martin Zenke, Erik A. L. Biessen

**Affiliations:** 1Institute of Innate Immunity, University of Bonn, 53127 Bonn, Germany; christa@uni-bonn.de; 2Department of Pathology, Cardiovascular Research Institute Maastricht (CARIM), Maastricht University, 6229 HX Maastricht, The Netherlands; pieter.goossens@maastrichtuniversity.nl (P.G.G.); erwin.wijnands@mumc.nl (E.W.); han.jin@scilifelab.se (H.J.); bart.legein@maastrichtuniversity.nl (B.L.); 3Central Diagnostic Laboratory, Maastricht University Medical Center, 6229 HX Maastricht, The Netherlands; t.oth@maastrichtuniversity.nl (T.O.); joris.vanderlocht@outlook.com (J.V.); 4Department of Biochemistry, Cardiovascular Research Institute Maastricht (CARIM), Maastricht University, 6229 HX Maastricht, The Netherlands; a.isaacs@maastrichtuniversity.nl (A.I.); mstoll@uni-muenster.de (M.S.); 5Amsterdam Medical Center, 1105 AZ Amsterdam, The Netherlands; e.lutgens@amsterdamumc.nl (E.L.); m.j.daemen@amsterdamumc.nl (M.J.A.P.D.); 6Department of Hematology, Oncology and Stem Cell Transplantation, RWTH Aachen University Medical School, 52074 Aachen, Germany; martin.zenke@rwth-aachen.de; 7Helmholtz Institute for Biomedical Engineering, RWTH Aachen University, 52074 Aachen, Germany; 8Institute for Molecular Cardiovascular Research (IMCAR), RWTH Aachen University, 52074 Aachen, Germany

**Keywords:** dendritic cells, cholesterol, toll like receptors, viral response, nuclear receptor, interferon

## Abstract

Background: Pathogens or trauma-derived danger signals induced maturation and activation of plasmacytoid dendritic cells (pDCs) is a pivotal step in pDC-dependent host defense. Exposure of pDC to cardiometabolic disease-associated lipids and proteins may well influence critical signaling pathways, thereby compromising immune responses against endogenous, bacterial and viral pathogens. In this study, we have addressed if hyperlipidemia impacts human pDC activation, cytokine response and capacity to prime CD4^+^ T cells. METHODS AND RESULTS: We show that exposure to pro-atherogenic oxidized low-density lipoproteins (oxLDL) led to pDC lipid accumulation, which in turn ablated a Toll-like receptor (TLR) 7 and 9 dependent up-regulation of pDC maturation markers CD40, CD83, CD86 and HLA-DR. Moreover, oxLDL dampened TLR9 activation induced the production of pro-inflammatory cytokines in a NUR77/IRF7 dependent manner and impaired the capacity of pDCs to prime and polarize CD4^+^ T helper (Th) cells. CONCLUSION: Our findings reveal profound effects of dyslipidemia on pDC responses to pathogen-derived signals.

## 1. Introduction

Plasmacytoid dendritic cells (pDCs) are instrumental in host defense against viral and microbial infections. Toll-like receptor (TLR) 7 and 9 activation of pDCs [1] will induce their maturation as well as the release of large amounts of type I interferons (IFNs; IFN-α, -β, -λ, -ω). These IFNs will tune Th1/2 differentiation, conventional DC (cDC) maturation, and stimulation of B cells to differentiate into antibody secreting plasma cells [1,2]. In addition to their role in infection response, pDCs are also seen to be implicated in the pathophysiology of a number of autoimmune diseases, such as systemic lupus erythematosus (SLE) or psoriasis, through uncontrolled production of type I IFNs able to stimulate auto-reactive T and B cells [3], and to contribute to peripheral T cell tolerance, as has been shown in cardiac transplantation and oral tolerance studies [4,5]. Third, we and others have demonstrated proinflammatory as well as tolerogenic roles for pDCs in chronic inflammatory diseases such as atherosclerosis, depending on the model and setting [6,7,8].

Tissue-resident pDCs are exposed to danger signals present in a microenvironment of interstitial fluid, lymph, and (leaked) plasma. The local context will likely tone pDC-mediated immune responses. In the setting of atherosclerosis, extravasated (oxidized) lipoproteins, laden with bioactive lipids, may be particularly potent context contained modifiers. Conditions of hyperlipidemia or oxidative lipoprotein modification may therefore directly compromise DC function [9,10]. Hypercholesterolemia was indeed seen to impair DC migration [11], while leaving antigen-presentation and T cell priming ability by several DC subsets unaffected [12,13]. At a functional level, hyperlipidemia was recently reported to affect murine CD8^−^ conventional DCs (cDCs), attenuating their response to TLR stimulation and host resistance to Leishmania major [14]. Oxidized phospholipids were shown to act as being immuno-inhibitory on LPS-induced DC maturation and cytokine release [15]. However, no studies thus far have addressed the effects of hyperlipidemia on pDC-mediated immune responses. 

In this study we therefore aimed to investigate the effect of hyperlipidemia on human pDC activation and CD4^+^ T cell polarization capability in vitro. We have shown here that exposure to modified lipoproteins dampens pDC maturation and functionality, thus blunting pDC-mediated immune responses. 

## 2. Material and Methods

### 2.1. Isolation and Culture of pDCs

Peripheral blood mononuclear cells (PBMCs) were prepared from heparinized whole blood of healthy donors by standard Ficoll–Paque (Sigma-Aldrich, Seelze, Germany) density gradient centrifugation. PDCs were isolated from PBMCs using the BDCA-4 Cell Isolation Kit or the pDC Diamond Kit (Miltenyi Biotec, Bergisch-Gladbach, Germany). Labelled cells were enriched using permanent magnets yielding a high-gradient magnetic field (midiMACS separators, Miltenyi Biotec). The positively sorted pDC fraction was analyzed by flow cytometry on a BD Canto II (BD Bioscience, Bedford, MA, USA) according to their BDCA-2^+^, CD123^+^, CD11c^−^, CD3^−^, CD14^−^, CD19^−^, CD20^−^ and CD56^−^ phenotype and reached > 90% purity. Isolated pDCs were seeded in round-bottomed 96-well cell culture plates at a density of 5 × 10^4^ cells/ well in AIM-V medium (Invitrogen, Darmstadt, Germany), and cultured for 24 h at 37 °C and 5% CO_2_ in the presence of 2.5 µM CpG-A (ODN2216; Invivogen, San Diego, CA, USA), with or without LDL (50 µg/mL) or oxLDL (10 µg/mL or 50 µg/mL) and plain medium supplemented with LDL (50 µg/mL) or oxLDL (10 µg/mL or 50 µg/mL). In similar control experiments pDCs were stimulated by 5 µg/mL imiquimod (IMI; Invivogen, San Diego, CA, USA); 10 ng/mL lipopolysaccharide (LPS from *E. coli*, O55:B5, Merck, Kenilworth, NJ, USA) or 10 µg/mL of anti-CD40 antibody clone G28.5 (Bioceros, Utrecht, The Netherlands), instead of CpG-A.

### 2.2. OxLDL Preparation

Low-density lipoprotein (LDL) (1.006 ≤ d ≤ 1.063 g/mL) was isolated from human plasma of normolipidemic healthy individuals by ultracentrifugation. The plasma density was raised to 1.055 g/mL using KBr. After ultracentrifugation at 32,000 rpm at 4 °C for 16 h using a SW41Ti rotor Beckman Instrument, the light fraction, containing LDL, was collected and stored under nitrogen. LDL concentration was determined using the bicinchoninic acid (BCA) protein kit (Thermo Scientific, Rockford, IL, USA). LDL concentration was adjusted to 0.5 mg/mL of protein by dilution in PBS. Cu^2+^-mediated oxidation was conducted at 37 °C for 24 h against 0.32 mM CuSO_4_/PBS. The reaction was stopped by addition of 50 µM ethylene-diamid-tetraacetate (EDTA) and extensive dialysis at 4 °C against PBS containing 0.5 M EDTA for 24 h. LDL or modified LDL were tested for endotoxin using a Limulus Amebocyte Lysate (LAL) assay (Hycult Biotechnology, Uden, The Netherlands), according to the manufacturer’s instructions. Endotoxin concentration of all samples was below 5 pg/mL.

### 2.3. Labeling of Lipoproteins and CpG

LDL and oxLDL were labeled with the fluorescent lipid 1,1′-dioctadecyl-3,3,3′,3′-tetramethylindocarbocyanine perchlorate (DiI; Invitrogen, Darmstadt, Germany), by incubating 3 mg/mL DiI per ml of LDL or oxLDL, overnight and in the dark, while continuously stirring in a 37 °C water bath. Free DiI was removed by size exclusion chromatography using PD-10 columns (GE Healthcare, Uppsala, Sweden). DiI-labeled LDL and oxLDL uptake was determined by flow cytometry on a BD Canto II (BD Bioscience, Bedford, MA, USA). Ready-to-use AlexaFluor 488-labelled CpG-ODN 2216 (type A) was ordered via Molecular Probes/Invitrogen.

### 2.4. Oil-Red-O Staining

In total, 5 × 10^4^ pDCs were cytospun at 700 rpm for 5 min on glass slides. After drying the slides for 1 h, sections were incubated in PBS for 5 min, dipped 10× in 60% 2-propanol and stained with Oil-Red-O solution for 20 min. After dipping 10× in 60% 2-propanol, sections were cleared under tap water. Cell nuclei were stained with hematoxylin. Sections were mounted with IMSOL and covered.

### 2.5. Flow Cytometry

Isolated pDCs were stained with fluorescein isothiocyanate (FITC)-conjugated CD3 (1:100, DAKO, Glostrup, Denmark), CD14 (1:100, eBioscience, San Diego, CA, USA), CD16 (1:100, DAKO), CD19 (1:100, DAKO), CD56 (1:100, eBioscience), CCR7 (1:20, BD Bioscience, Bedford, MA, USA) and CD40 (1:100, BD Bioscience), R-phycoerithrin (PE)-conjugated CD86 (1:500, BD Bioscience) and CD83 (1:20, BD Bioscience), PerCP-Cy5.5-conjugated CD123 (1:50, BD Bioscience), allophycocyanin (APC)-conjugated BDCA-2 (1:50, Miltenyi Biotec, Bergisch-Gladbach, Germany), phycoerithrin-Cyanine7 (PE-Cy7)-conjugated CD11c (1:20, eBioscience), allophycocyanin-H7 (APC-H7)-conjugated HLA-DR (BD) and Pacific Blue-conjugated CD4 (1:20, eBioscience) monoclonal antibody (mAb). Life-dead cell staining was accomplished using 7-AAD, a marker that is intercalating in non-vital cell DNA. All flow cytometry analyses were performed on a BD Canto II (BD Bioscience).

### 2.6. Mixed Leukocyte Reaction

Autologous naïve CD4^+^ T cells were isolated from pDC-depleted PBMCs by use of the human naive CD4^+^ T Cell Isolation Kit (Miltenyi-Biotec, Bergisch-Gladbach, Germany). As determined by staining with V500-conjugated CD3 mAb, APC-H7-conjugated CD4 mAb, FITC-conjugated CD8 mAb, PE-conjugated CD45RO mAb, V450-conjugated CD45RA mAb and APC-conjugated CD25 mAb (BD Bioscience, Bedford, MA, USA), populations containing 99.9% CD3^+^ CD4^+^ CD45RA^+^ CD45RO^−^ CD8^−^ T cells were routinely obtained. For the mixed leukocyte reaction (MLR), autologous purified naïve CD4^+^ T cells (5 × 10^5^ cells/ well) were incubated in 96-well cell culture plates with 1 × 10^5^ pDCs, pre-cultured for 24 h as described above and pre-incubated 1 h before adding the T cells with Staphylococcus Enterotoxin B (SEB; 10 ng/mL). MLRs were cultured for 7 days. The assay was performed in triplicate.

### 2.7. Cytokine Measurement

PDCs were treated as indicated, and cytokine concentrations of IFN-α, IL-6, IL-12p70, IL-10, TNF-α, RANTES (CCL5), MIP-1α (CCL3), MIP-1β (CCL4) and CXCL10 (IP10) in cell-free supernatants were measured after 24 h using the Cytometric Bead Array (CBA) human soluble protein kit (BD Bioscience, Bedford, MA, USA), according to the manufacturer’s instructions. For the T cell polarization assay, mixed lymphocyte reaction (MLR) supernatants were harvested at day one to seven of co-culture and analyzed for IFN-γ, TNF-α (T helper 1 cytokines), IL-4, IL-5, IL-13 (T helper 2 cytokines), IL-17A (T helper 17 cytokine), and IL-10 (Treg cytokine) by CBA kit. ELISA plates from Greiner Bio-One (Alphen a/d Rijn, The Netherlands) were used.

### 2.8. RNA Isolation 

Human pDCs were isolated as described above and purity was checked by flow cytometry (>90%). PDCs (approx. 100,000 cells, *n* = 3) were challenged overnight with CpG-A (2.5–5 µM) in the absence or presence of oxLDL, after which the cells were washed and suspended in DEPC RNase-free water. Unstimulated pDCs were used as controls. Total RNA was extracted from the cell suspensions using the RNeasy Micro kit (Qiagen, Hilden, Germany), according to the manufacturer’s protocol for extraction of RNA from microdissected cryosections. DNA was removed by using 50U DNase I (Qiagen). RNA quantity and quality were determined using the NanoDrop-1000 Spectrophotometer and the Agilent’s 2100 Bioanalyzer System (Wilmington, DE, USA; Agilent Technologies, Palo Alto, CA, USA); average yield was 115 ng RNA; RNA integrity numbers were >6.5. 

### 2.9. Quantitative Real-Time PCR (qRT-PCR) Analysis

For cDNA synthesis, total RNA was subjected to reverse transcription using the iScript cDNA Synthesis Kit (Biorad, Hercules, CA, USA). qRT-PCR was performed using SYBR Green PCR Master Mix (Biorad) as described previously. IFN-α delta-Ct expression is presented relative to beta-actin (housekeeping gene) expression. 

### 2.10. Microarray Analysis

Labeled cRNA was prepared and used on the Illumina HT12 v4.0 Beadchip array for hybridization. Hybridized chips were scanned by Illumina BeadStation (Illumina, Inc., San Diego, CA, USA). Raw image analysis and signal extraction was performed with Illumina Beadstudio Gene Expression software with default settings (no background subtraction) and no normalization. Raw and normalized microarray data are accessible in the Gene Expression Omnibus (GEO) database through the accession GSE133902. Using R package Limma (v3.48.0), expression data were analyzed for differentially expressed genes (DEGs, mRNA isoforms of the same gene were averaged) following background correction and normalization. For CpG-driven effects, i.e., CpG vs. control and oxLDL + CpG vs. oxLDL, thresholds were set as log_2_ fold change > 1 or <−1, with Benjamini–Hochberg adjusted *p*-value < 0.05 for significance; for oxLDL-driven effects, i.e., oxLDL vs. control and oxLDL + CpG vs. CpG, thresholds were set as log_2_ fold change > 1 or <−1, and *p*-value < 0.05 for significance. OxLDL-induced DEGs upon CpG treatment (oxLDL + CpG vs. CpG) were analyzed for associated biological pathways by gene set overrepresentation analysis (GSOA) and gene set enrichment analysis (GSEA) using R package clusterProfiler [16] (v4.0.0) and ReactomePA [17] (v1.36.0) based on the Reactome pathway database, with Benjamini–Hochberg adjusted *p*-value < 0.05 as significance. In addition, DEGs of oxLDL+ CpG vs. CpG were further analyzed for motif enrichment by R package RcisTarget [18] (v1.12.0) using the database file hg38__refseq-r80__10kb_up_and_down_tss.mc9nr.feather (24,453 motifs, 2993 tracks, 10 kb around the transcription start site). Protein–protein interaction networks were constructed for the top differentially expressed mRNA (confidence level 0.4, evidence based, with exclusion of disconnected nodes). Graphical visualization was conducted via Cytoscape. 

### 2.11. Nuclear Localization Studies

The pDC were isolated, enriched and plated as described above. Cells were stimulated in triplicate for 5 h with oxLDL (25 μg/mL) in the presence of human IL-3 (10 ng/mL Peprotech) with or without Nur77 antagonist (20 μM; TMPA; ThermoFisher/Calbiochem) or with the Nur77 agonist lithocholic acid (20 μM; LCA; Sigma, St. Louis, MO, USA), followed by CpG-A activation of TLR9 for 1 h (2.5 μM, Invivogen, San Diego, CA, USA). They were fixed with PFA and permeabilised with 0.1% Triton X100 prior to staining with anti-Nur77 (Alexa Fluor 555; Bioss), anti-phospho-IRF7 (Ser471 + Ser472; Alexa Fluor 488; Bioss, Woburn, MA, USA) and Hoechst 33342 (Sigma) in a 96-wells plate (V-shaped; Greiner). Stained cells were spotted on glass slides using a Cytospin centrifuge (Miles Scientific, Newark, DE, USA) and mounted with ProLong^TM^ Gold (ThermoFisher, Waltham, MA, USA). Ten randomly selected fields of views were imaged for each condition using a confocal microscope (Leica TCS SP8). The mean nuclear and cytoplasmic fluorescence for each channel was quantified (approx. 20 cells per field) using ImageJ and corrected for background signal.

### 2.12. NF-κB Immunofluorescent Staining 

Isolated pDCs were incubated in AIM-V medium supplemented with LDL (25 μg/mL) or oxLDL (25 μg/mL) in the presence of human IL-3 (10 ng/mL Peprotech) for 6 h, accompanied by CpG-A stimulation overnight (2.5 μM, Invivogen). After stimulation, cells were washed twice with phosphate-buffered saline (PBS), transferred into v-shaped 96-well plates, and spun at 1600 rpm for 3 min. Subsequently, cells were fixed in 2% paraformaldehyde (PFA) for 30 min at room temperature. After spinning down (2000 rpm, 3 min), cells were permeabilized in 90% ice-cold methanol for 30 min on ice, spun down (2000 rpm, 3 min) and resuspended in PBS/2 mM EDTA buffer. Cells were incubated for 60 min on ice with anti-p65 primary antibody (dilution 1/1600; Cell Signaling). After washing twice in PBS/2 mM EDTA cells were incubated for 30 min in the dark on ice with goat-anti mouse AlexaFluor647 secondary antibody (dilution 1/1000; Molecular Probes/Invitrogen). Cell nuclei were counterstained with Dapi. After, pDCs were cytospun at 700 rpm for 5 min on glass slides and analyzed by Leica SP5 confocal microscope.

### 2.13. Statistical Analysis

Data were analyzed with the use of Graph Pad Prism 5 (Graph Pad Software Inc., San Diego, CA, USA). All data are expressed as mean ± SEM (unless otherwise mentioned). Group comparisons were conducted by one way ANOVA (with Bonferroni post-testing). To compare individual treatment groups, we used unpaired 2-tailed Student’s *t*-test; nonparametric data were analyzed using a Mann–Whitney *U*-test. Values of *p* < 0.05 were considered statistically significant.

## 3. Results 

### 3.1. Human pDCs Are Able to Engulf Lipoproteins

First, we examined whether human PBMC-derived pDCs are able to internalize oxLDL. The effective isolation of immature pDCs to ±90% purity was confirmed by flow cytometry for surface marker expression of BDCA-2 and CD123 (Appendix A). Immature, as well as CpG-A matured, pDCs were incubated for 24 h in the absence or presence of DiI-labelled oxLDL (10 and 50 μg/mL). Cells exhibited modest lipoprotein uptake, as assessed by flow cytometry (*p* < 0.05). CpG-stimulation markedly augmented DiI-oxLDL uptake by pDCs, in particular at the highest oxLDL concentration (*p* < 0.001; Appendix A). Lipid accumulation by pDCs exposed to DiI-labelled oxLDL (and to a lesser extent also DiI labelled LDL) was confirmed by fluorescence microscopy both at baseline and after CpG activation (Appendix A), where we observed a characteristic punctate presence of Oil-Red-O stained vacuoles in (ox)LDL exposed pDCs. Pilot experiments suggested that internalized CpG-A tended to colocalize with intracellular LDL, but only partly with oxLDL pools (Appendix A), although this needs quantitative confirmation. Overnight incubation of immature pDCs with oxLDL did not lead to increased cell death at concentrations of up to 50 µg/mL (Appendix A), and average cell viability after 24 h amounted 25–30%. While pDCs stimulated for 24 h with CpG-A showed much less cell death (10–15%), cell viability remained unaffected by co-incubation with oxLDL (10–50 μg/mL) (Appendix A). 

### 3.2. (Oxidized) Lipoprotein Dampens CpG-A Induced Up-Regulation of Maturation Markers 

To address the effects of oxLDL accumulation on pDC maturation, we incubated human pDCs for 24 h with oxLDL (0, 10 and 50 μg/mL) in the absence or presence of CpG-A (2.5 µM). CpG-A activated pDCs acquired a characteristically mature phenotype compared to non-stimulated controls, as demonstrated by a significantly increased expression of maturation markers, including CD83, CD86, HLA-DR and the lymphoid-homing receptor CCR7 (Figure 1A,B). BDCA-2, a pDC specific type II C-type lectin [19], is highly expressed in immature pDCs, but down-regulated in mature pDCs, rendering it a useful marker for immature pDCs. As expected, CpG-stimulated pDCs showed significantly decreased BDCA-2 expression (*p* < 0.001; Figure 1A). OxLDL pretreatment quenched the CpG-A induced surface expression of CD83, CD86, HLA-DR and CCR7 in a dose-dependent manner, to levels comparable to that of immature pDCs (Figure 1A). In contrast, BDCA-2 surface expression remained unexpectedly low in CpG-A/oxLDL co-stimulated pDCs (Figure 1A). Bioactive oxidized lipids, such as oxidized phospholipids contained in oxLDL, can profoundly impact the course of an immune response [20], as has been reported for pDC activation, amongst others. To dissect the contribution of oxidized lipid constituents we assessed the effects of LDL (50 µg/mL) on pDC maturation in a similar setup. Treatment with LDL did not influence pDC maturation status. However, like oxLDL, LDL also abrogated CpG-induced pDC maturation as indicated by the significant down-regulation of CD83, CD86, HLA-DR and CCR7, as well as by the concomitant up-regulation of BDCA-2 (Figure 1C).

### 3.3. Lipoprotein Pre-Treatment Inhibits TLR7 and TLR9-Induced pDC Maturation

Next, we investigated whether lipoproteins inhibit CpG-A effects by preventing CpG-A uptake, or by shielding CpG-A to interact with TLRs. Pre-treatment of pDCs with LDL or oxLDL prevented CpG-induced up-regulation of maturation markers, including CD40, CD86, HLA-DR, and CCR7 (Figure 2A). BDCA-2 surface expression remained up-regulated in CpG-stimulated pDCs pre-treated with lipoproteins. In contrast, CpG-induced pDC activation was not affected by post-treatment with LDL or oxLDL. Cells displayed elevated expression levels of the aforementioned pDC maturation markers (Figure 2A) and had increased IFN-α and TNFα production (Appendix A). To conclude, pre-incubation of pDCs with LDL or oxLDL ablates TLR9-induced activation by CpG.

PDCs typically express TLR7 and TLR9, which recognize single-stranded RNA and double-stranded DNA, respectively [21]. The observation that oxLDL interferes with CpG-A induced TLR9 signaling in pDCs led us to investigate whether oxLDL also dampened TLR7 activation of pDCs. In the same experimental setup, we measured imiquimod (TLR7) and LPS (TLR4) responses in the absence and presence of oxLDL (50 µg/mL). We observed similarly strong effects on CD86, CD83, HLD-DR, CCR7 and BDCA2 expression upon imiquimod as compared to CpG-A challenge (Figure 2B). Analogous to CpG-A stimulation, oxLDL was also able to quench the imiquimod responses. LPS challenge only led to minor changes in pDC marker expression, which were neither affected by LDL nor oxLDL treatment (Figure 2B). 

The LDL and oxLDL associated inhibition of TLR activation induced pDC maturation led us to investigate whether pDC cytokine production was also altered. In the context of systemic viral or bacterial infections, pDCs rapidly release large amounts of type I IFNs as well as other pro-inflammatory cyto- and chemokines, which promote lymphocyte recruitment to sites of infection [22,23,24,25]. Here, co-treatment of immature pDCs with LDL or oxLDL (both 50 µg/mL) and CpG-A for 24 h almost completely reversed the CpG-induced production of pro-inflammatory cytokines such as IFN-α, TNF-α and IL-6 (*p* < 0.001; Figure 3A). Additionally, CpG-A induced release of certain chemokines including CCL3, CCL4 and CXCL10 was nearly completely ablated, upon exposure to (oxidized) lipoproteins (*p* < 0.001; Figure 3B). Concentrations of IL-10, IL-12p70, and CCL5 were undetectable under all conditions tested (data not shown). Altogether, our data indicate that exposure to (modified) lipoproteins per se does not induce in vitro maturation of human pDCs; rather, it appears to quench CpG-A induced pDC activation.

### 3.4. OxLDL Treatment Suppresses TLR9-Induced pDC Activation and T Cell Stimulatory Capacity

Next, we analyzed the impact of oxLDL exposure on the T cell stimulatory capacity of mature, CpG-A activated pDCs by autologous mixed leukocyte reactions. Non-activated pDCs were used as control. As expected, CpG-A activated pDCs augmented Th1 polarization of naïve CD4^+^ T cells after 7 days of co-culture, as judged by the markedly increased IFN-γ and TNF-α cytokine release (Figure 4A). Similarly, CpG-A treated pDCs had augmented IL-13 production on day 7 of co-culture, whereas the production of prototype Th2 cytokines IL5 and IL4 remained unaffected (Figure 4B). Both Th1 and Th2 stimulatory capacity of CpG-stimulated pDCs was effectively blunted by co-treatment with oxLDL (Figure 4A,B), consistent with the impaired maturation and activation capacity of oxLDL treated pDC. Of note, pDC exposed T-cells did not produce detectable levels of IL-17A in all conditions (Figure 4C), while IL-10 cytokine concentrations were significantly up-regulated on day 7 of co-culture with immature as well as matured pDCs, an effect that was quenched by oxLDL (Figure 4D). 

### 3.5. Involvement of the NUR77/TRAF6 Axis in oxLDL Inhibition 

Next, we sought to dissect the underlying mode of action of oxLDL inhibitory functions in pDCs by transcriptional profiling (for workflow see Figure 5A). As expected CpG-A activation led to a sharp dysregulation of pDC gene expression profiles, with 2420 up- and 2948 downregulated genes (FDR corrected *p* value: 0.05; Figure 5B); oxLDL was partly able to blunt this effect, as judged by hierarchical clustering based on the most differentially expressed genes (cutoff *p* < 0.005; Figure 5B,C). Intersection analysis revealed a significant overlap of oxLDL effects on immature versus mature pDCs (*p* = 2.10^−32^), but also highlighted a considerable extend of state-specific effects of oxLDL (Appendix A). Based on the top differential mRNA of CpG-A treated pDCs +/− oxLDL we constructed a gene ontology functional network by BiNGO and Enrichment Map, indicating clear down-regulation of type-I IFN immune responses and STAT activation, and up-regulation of ficolin-1-rich granulae and nucleoplasm genes. This was essentially confirmed by TopGo analysis (Appendix A). We further analyzed the Met-Flow panel [26]. As shown in Appendix A, TLR9 activation (CpG) led to downregulation of phosphate metabolism (SLC20A1) and glucose uptake (SLC2A1 = GLUT1), respectively, the main glucose uptake transporters in pDCs with a minor increase in ASS1 (arginosuccinate synthesis). No effects were seen on fatty acid synthesis (ACACA), fatty acid oxidation (CPT1A), oxidative phosphorylation (ATP5F1), oxidative stress (PRDX2), glycolysis (HK1, G6P) and PPP (G6P). This suggests that metabolic regulation in human pDCs may differ from that in mouse pDCs, as recently reported for macrophages [27]. While oxLDL exposure did not affect synthesis, it did further decrease oxPHOS activity (ATP5F1) and tended to increase IDH2 (Kreb’s cycle) in CpG stimulated cells (Appendix A).

Zooming in on the downstream pathways, EnrichR revealed a highly significant enrichment of TRAF6-mediated IRF7 signaling among the oxLDL differentially expressed pDC genes (Figure 5E,F); the relevance of the TRAF6-IRF7 axis was also highlighted in the strong protein–protein interaction network that could be constructed based on the set of top differentially expressed genes (Appendix A). To address whether this axis was interfering at a transcriptional level we conducted a motif search analysis. However, this failed to provide further support for the relevance of this axis (data not shown), suggesting a post-transcriptional mode of action. 

As NUR77 has both been implicated in oxLDL responses, as well as in TRAF6 signaling, we zoomed in on this orphan receptor as an intracellular (ox)LDL sensor. PDC stimulation assays with combined confocal microscopy analysis revealed that CpG-A induced NUR77 nuclear translocation was blunted by oxLDL pre-treatment. Nuclear translocation of phospho-IRF7, a key step in type I IFN response, essentially mirrored this pattern (Figure 6A–C). Importantly, the NUR77 agonist lithocholic acid (LCA) could emulate oxLDL’s inhibitory effects on Nur77 and phosphor-IRF7 (*p* < 0.001), whereas ethyl-2-(2,3,4-trimethoxy-6-(1-octanoyl) phenyl) acetate (TMPA), a NUR77 antagonist, was able to rescue the inhibitory effect of oxLDL on CpG nuclear translocation (Figure 6A–C). Likewise, CpG-A induced NF-kB nuclear translocation was blunted by oxLDL pre-treatment (data not shown), which was reflected on the gene expression level by a significantly reduced expression of IFN-α (Appendix A). In line, TNF-α levels (pg/mL), measured in cell supernatants, were markedly decreased in oxLDL/ CpG-A compared to CpG-A treated pDCs (Appendix A).

## 4. Discussion

Hyperlipidemia has been shown to impair host immune responses not only against yeast and bacterial but also against viral infections [30,31,32,33]. However, the underlying mechanisms are not well understood. As pDCs are instrumental in anti-viral defense by producing large amounts of type I IFNs and presenting endogenous antigens to naïve T cells, this renders them a plausible target for hyperlipidemia associated impairment of viral responses [34,35], Here, we demonstrate that human pDCs are hypo-responsive to maturation triggers after oxLDL exposure, and fail to mount effective Th cell polarization in vitro. Moreover, we were able to pinpoint this impairment to the activation of NUR77 dependent suppression of the IRF7/TRAF6 pathway. These effects might have striking implications for pDC-mediated pathogen as well as general immune responses under pathological conditions associated with dyslipidemia, such as atherosclerosis or obesity.

As we show, human pDCs can take up and accumulate lipoproteins, which aligns with the reported ability of murine pDCs to ingest oxLDL [7]. Lipoprotein (LDL or oxLDL) accumulation dose-dependently dampened the CpG-induced IFN and cytokine responses. Additionally, the hyperlipidemic environment abrogated the up-regulation of co-stimulatory molecules, such as CD40, CD83 and CD86, as well as HLA-DR, markers which are pivotal for T cell priming [36]. CD4^+^ T cell polarization was, as a result, completely quenched. Interestingly, lipid accumulation after oxLDL and LDL components are essentially similar, hinting to non-scavenger receptor mediated uptake. As both oxLDL and, albeit to a lesser extent, LDL, are capable of suppressing CpG activation, this suggests a common actor contained in both lipoproteins to be responsible for this effect.

OxLDL exposure was accompanied by a markedly reduced surface expression of the lymphoid-homing receptor CCR7 in CpG-stimulated pDCs. Down-regulation of this receptor may limit pDC migration from injured tissue to secondary lymphoid organs, thereby dampening the induction of effective T cell responses in vivo [37,38]. Next to CCR7 several other chemotactic cues were reported to control pDC trafficking under baseline (L-selectin, CXCR4 and Chem23) [39], and inflammatory conditions (CCR5 and CXCR3) []. It remains to be determined whether functions of CCR5, CXCR3 and in particular ChemR23, which controls pDC migration towards secondary lymphoid organs, especially under inflammatory conditions [40] are also compromised under hyperlipidemic conditions. As the (ox)LDL effect does not occur with post-treatment, this suggest that pDC are receptive to the (ox)LDL inhibitory effects when cells are exposed prior to or immediately after TLR activation; pointing to interference with the TLR signaling pathway itself.

Lipoprotein treatment dampened CpG-A induced production of cytokines such as IFN-α, but also IL-6 and TNF-α, as well as of the T cell attracting chemokines CCL3 and CCL4, identifying (ox)LDL as a potent suppressor of inflammatory cytokine and chemokine production in human pDCs [24,25]. This is in line with earlier findings by our group demonstrating a dampened CpG-induced IFN-α release by pDCs into the circulation in a Western type diet over chow fed low-density lipoprotein deficient (*Ldlr*^−/−^) mice. However, baseline IFN- levels in plasma were unaltered, in keeping with the fact that the bulk of IFN-a in plasma is derived from myeloid cells [6]. Compromised function, with lowered expression and secretion of IFN-α, has been observed before in circulating pDCs, isolated from patients with carotid endarterectomy [6], or with coronary artery disease [41]. However, we await future studies of pDC function and phenotype in patients with familial hypercholesterolemia.

Considering the mode of action for oxLDL-mediated TLR9 suppression, our studies could be explained by oxLDL induced interference with TLR-ligand interaction [15,42] and/or uncoupling of TLR signaling in pDCs, as has previously been reported for the dyslipidemia associated inhibition of murine CD8α^−^ cDC activation [14]. Although our preliminary data on macrophages suggest that oxLDL does not interfere with CpG uptake or its binding to TLR9, we cannot completely exclude such uncoupling to occur further downstream, at the level of vesicle fusion or TLR compartmentalization. A second, more likely option is that oxLDL entrapped lipid constituents tune TLR induced NFkB and/or IRF7 signaling. Our computational studies indeed supported such post-translational interference of oxLDL with the IRF7/TRAF6 axis. Further in vitro analysis identified Nur77 as a potential transducer of oxLDL mediated TLR suppression, as its inhibition rescued oxLDL effects, whereas its activation mirrored the oxLDL phenotype. Whether or not oxLDL also acts by binding to pDC surface receptors, such as BDCA-2, dendritic cell immuno-receptor, CXCL16 or immunoglobulin-like receptor 7, all capable of regulating type I IFN production, remains to be established [19,43,44].

Collectively, our data suggest that hyperlipidemia could induce immuno-tolerance in pDCs. PDCs acquire a tolerogenic phenotype by production of anti-inflammatory cytokines, such as IL-10 or TGF-β [45,46] or by indoleamine 2,3-dioxygenase [6], dampening immunogenic T cell responses and/or favoring Treg differentiation. However, we were unable to detect elevated levels of IL-10 or TGF-β in a pDC-conditioned medium after lipoprotein exposure, nor did lipoprotein treatment induce the generation of Tregs. Further studies will be required to firmly establish this point.

Taken together, our data show that a hyperlipidemic environment inhibits TLR7 and TLR9 dependent maturation and cytokine production by human pDCs and abrogates pDC-elicited T cell responses in vitro. This effect is mainly precipitated by NUR77 dependent suppression of the TRAF6/IRF7 axis, opening new avenues for rescue of compromised pDC functioning.

Given the extent of oxLDL suppression of pDC activation, it would be of utmost interest to investigate the influence of other plasma lipids (cholesterol, triglycerides) on the functionality of circulating human pDCs. While cholesterol itself has not been reported as Nur77 substrate, we indeed cannot exclude that cholesterol metabolites produced in lipid laden pDCs, rather than oxysterols entrapped in LDL, account for its activation.

Overall, it remains elusive whether hyperlipidemia renders pDCs functionally impaired and non-responsive to pathogen-derived signals in vivo in humans.

## Figures and Tables

**Figure 1 biomedicines-10-01152-f001:**
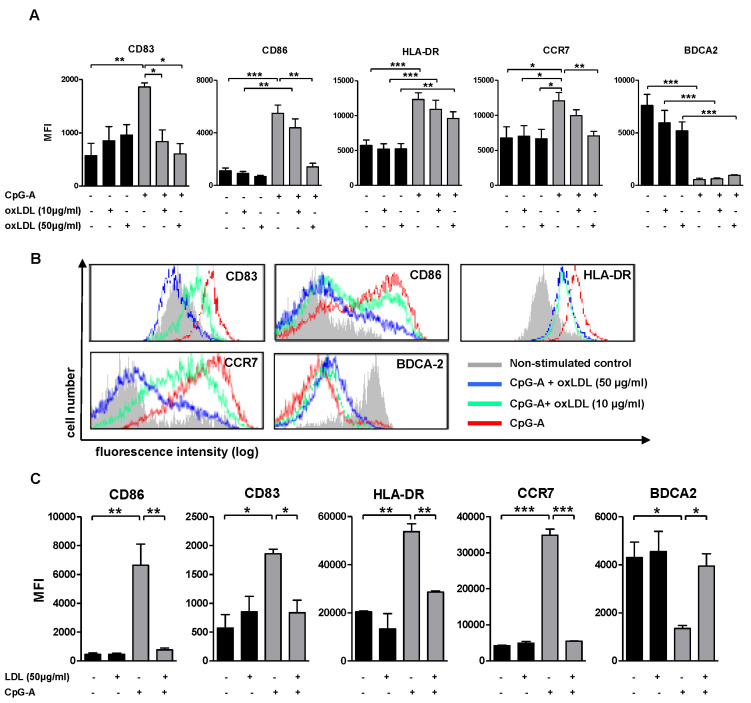
OxLDL inhibits CpG-induced expression of maturation markers by pDCs. Freshly isolated human pDCs were stimulated with 2.5 µM CpG-A in the absence or presence of lipoproteins; oxLDL and CpG-A were added simultaneously. Non-stimulated pDCs with or without oxLDL were used as control. After 24 h, cells were harvested, and the surface expression levels of the indicated markers were measured by flow cytometry. (**A**) Induction of pDC maturation markers (CD83, CD86, HLA-DR, CCR7, BDCA2) expressed as MFI. Exposure to oxLDL (10 and 50 µg/mL) dose-dependently reverses the CpG-A induced up-regulation of pDC maturation markers. (**B**) Maturation marker expression on non-stimulated pDCs (grey), CpG-A stimulated pDCs (2.5 µM; red) or CpG-A (2.5 µM) plus oxLDL stimulated pDCs (10 µg oxLDL/mL, green; 50 µg oxLDL/mL, blue). (**C**) Coincubation of pDCs with LDL (50 µg/mL) reverses CpG-A induced changes in pDC maturation marker expression. Data are shown as mean ± SEM of five independent experiments using different donor pDCs. * *p* < 0.05, ** *p* < 0.01, *** *p* < 0.001.

**Figure 2 biomedicines-10-01152-f002:**
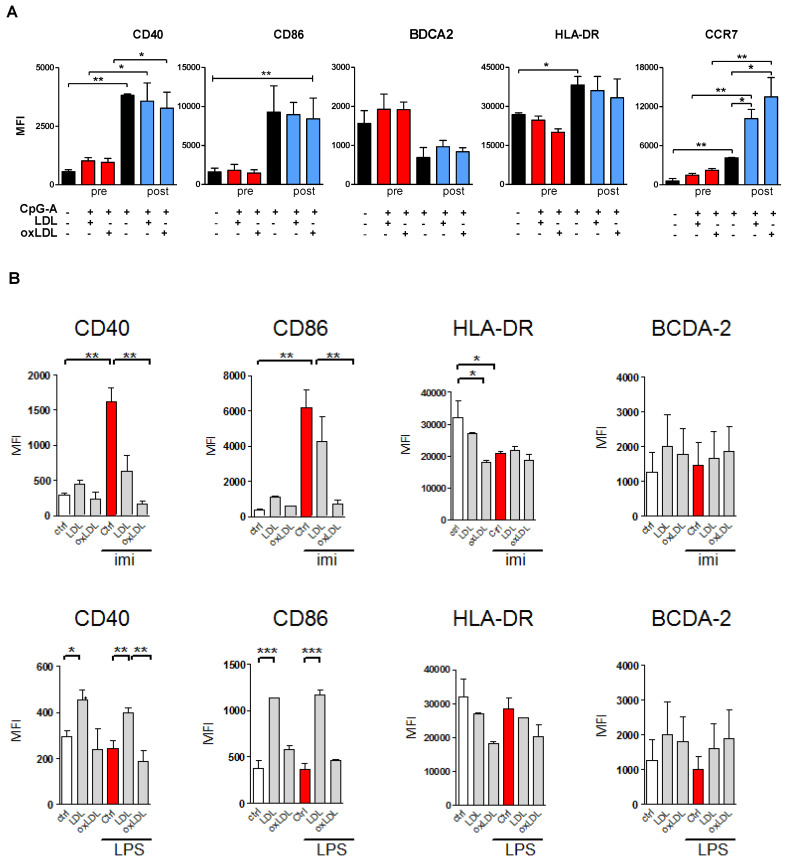
Prior lipoprotein treatment inhibits CpG-induced maturation. (**A**) Lipoprotein reversal of TLR induced pDC maturation. Freshly isolated human pDCs were stimulated for 4 h with (ox)LDL (50 µg/mL), and subsequently for 16 h with 2.5 µM CpG-A (red bars), or in reversed order (4 h of CpG-A + 16 h of (ox)LDL); blue bars). As controls, pDCs were stimulated with 2.5 µM CpG-A (positive control) or incubated in non-supplemented medium (negative control) (black bars). Subsequently, cells were harvested, and the surface expression levels of the indicated markers were measured by flow cytometry. Induction of maturation markers (CD40, CD86, HLA-DR, CCR7, BDCA-2) on pDCs are shown as MFI. (**B**) Co-treatment with (ox)LDL attenuates Imiquimod (red bars, 5 µg/mL) induced pDC activation, suggesting interference with TLR7 signaling as well. Unstimulated pDCs, in the absence or presence of LDL or oxLDL served as controls (black bars). LPS does not induce pDC maturation (lower panels). Data are shown as mean ± SEM of three independent experiments using different donor pDCs. * *p* < 0.05, ** *p* < 0.01, *** *p* < 0.001.

**Figure 3 biomedicines-10-01152-f003:**
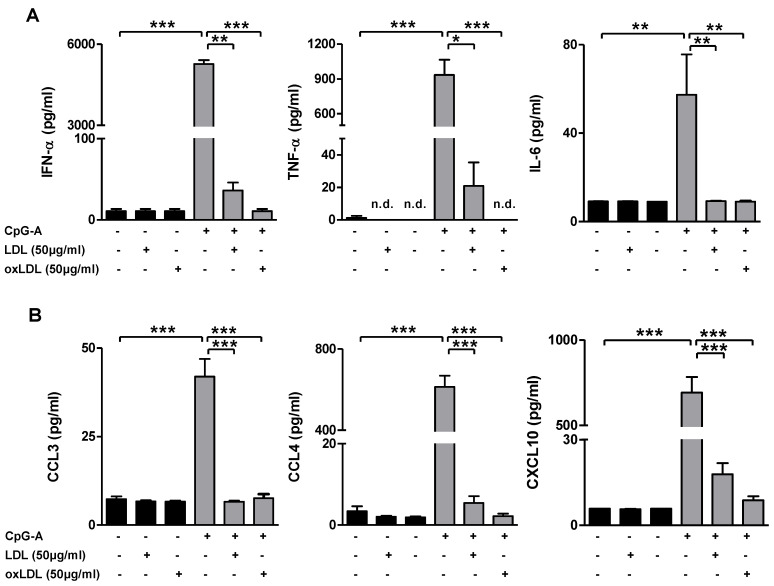
(Oxidized) LDL inhibits CpG-induced pro-inflammatory cytokine production by pDCs. Freshly isolated human pDCs were stimulated with 2.5 μM CpG-A either with or without LDL (50 µg/mL) or oxLDL (50 µg/mL). Non-stimulated pDCs were used as control. After 24 h, cell supernatants were analyzed for the indicated cytokines (**A**) and chemokines (**B**) by use of the CBA kit. Data are shown as mean ± SEM of three independent experiments using pDC preparations from different donors. * *p* < 0.05; ** *p* < 0.01, *** *p* < 0.001.

**Figure 4 biomedicines-10-01152-f004:**
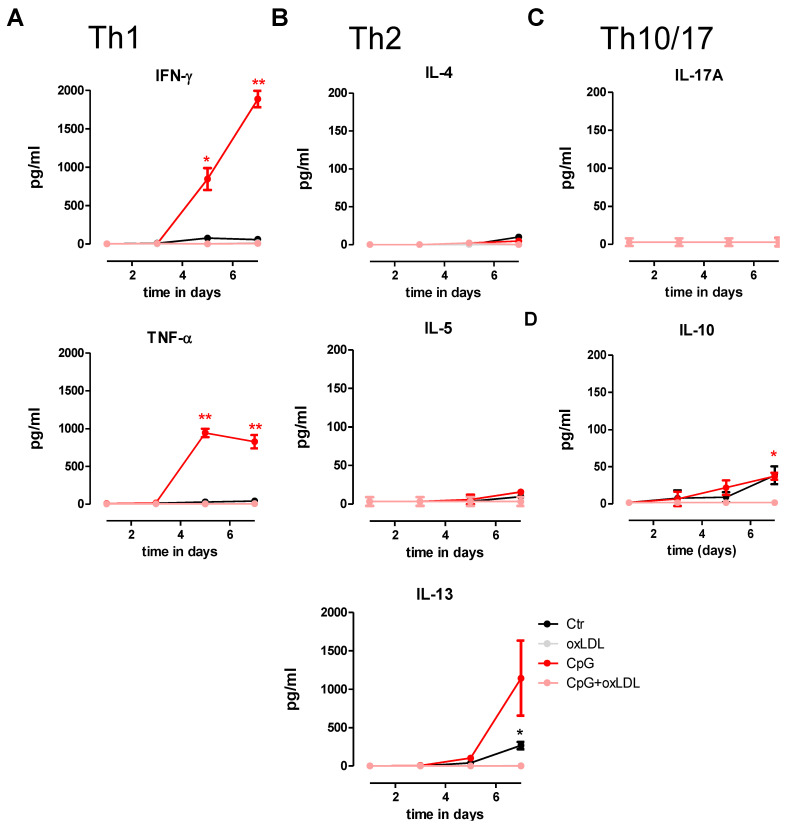
OxLDL skews T cell polarization of CpG-activated pDCs. Human autologous naïve CD4^+^ T cells were stimulated with immature CpG-A (2.5 µM) challenged pDCs, that had or had not been exposed to 25 µg/mL oxLDL, or with immature pDCs that had only been exposed to 25 µg/mL oxLDL. T cell polarization capacity (Th1, Th2, Th17 or Treg) was analyzed on day 1, 3, 5 and 7 of co-culture by measuring T cell cytokines in the cell supernatant (CBA kit). (**A**) Measurement of Th1 cytokines IFN-γ and TNF-α; (**B**) measurement of Th2 cytokines IL-4, IL-5 and IL-13; (**C**) measurement of Th17 cytokine IL-17A; (**D**) measurement of Treg cytokine IL-10. Data are shown as means of two independent experiments using different donor pDCs and T cells. * *p* < 0.05; ** *p* < 0.01 (color-coded according to treatment).

**Figure 5 biomedicines-10-01152-f005:**
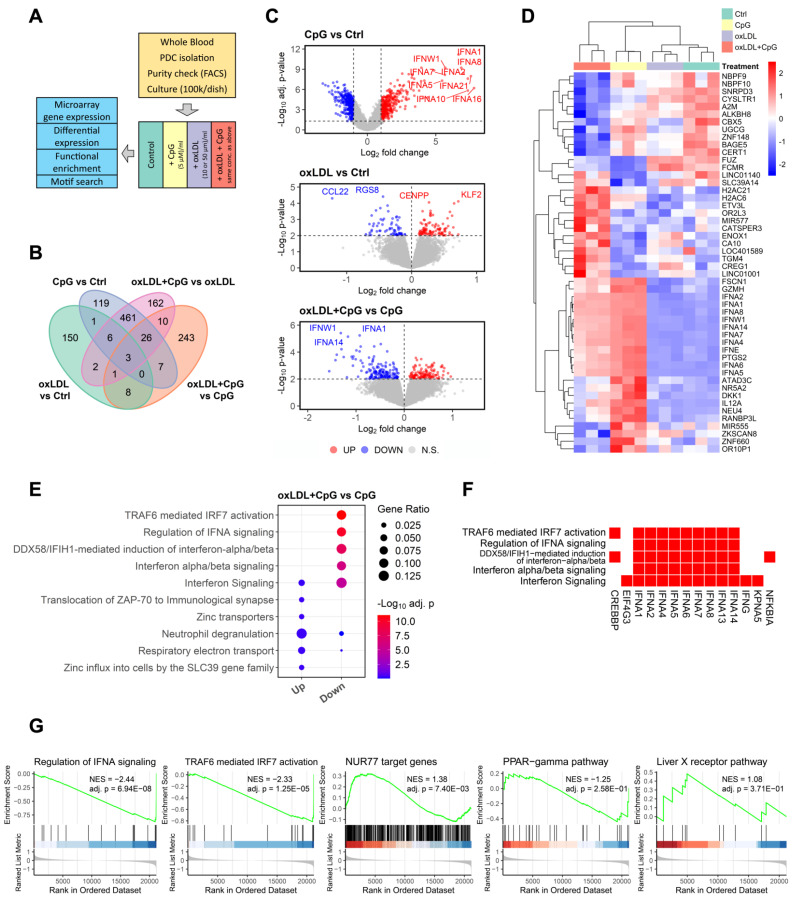
Transcriptional expression profiling reveals that oxLDL inhibits CpG-A induced activation by interfering with TRAF6 signaling and type-I interferon responses. (**A**) Flow scheme of the experimental setup of the profiling study. (**B**) DEGs of the four comparisons and their intersection. (**C**) Volcano plot of the gene expression data of CpG-A activated pDCs versus control (upper panel), oxLDL activated pDCs vs. control (middle panel), and CpG-A activated pDCs with/without oxLDL (lower panel). (**D**) Heatmap of gene expression of the 50 most significant oxLDL + CpG vs. CpG DEGs for the four study arms. (**E**) Gene set overrepresentation analysis of oxLDL + CpG vs. CpG DEGs. For either up-regulated or down-regulated DEGs, the five most significant reactome pathways are selectively shown. (**F**) Down-regulated oxLDL + CpG vs. CpG DEGs that are involved in the five Reactome pathways in (**E**). (**G**) Gene set enrichment analysis of oxLDL-induced overall gene expression upon CpG treatment (oxLDL + CpG vs. CpG, all genes). The two most significant Reactome pathways (first two), the 371 NUR77 targets retrieved from Harmonizome database [28], as well as two pathways from WikiPathways database [29] (last two) are selectively shown. For the comparison of oxLDL + CpG vs. CpG in (**E**–**G**), up-regulation, higher expression in oxLDL + CpG; down-regulation, higher expression in CpG.

**Figure 6 biomedicines-10-01152-f006:**
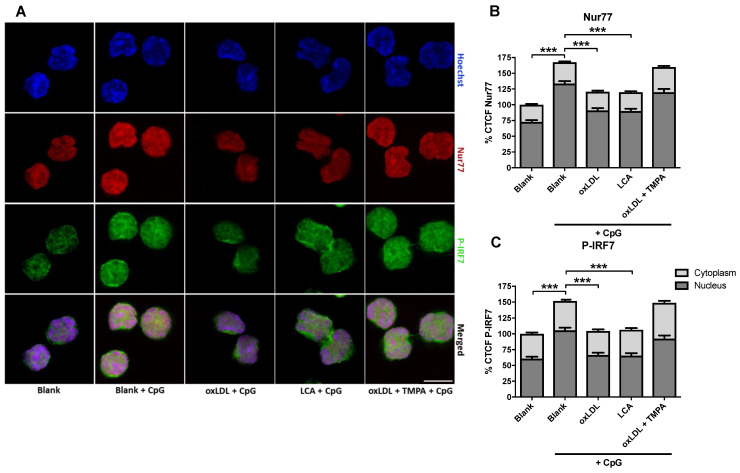
oxLDL (25 µg/mL), NUR77 agonist lithocholic acid (LCA, 20 µM) and/or NUR77 antagonist ethyl 2-[2,3,4-trimethoxy-6-(1-octanoyl)phenyl] acetate (TMPA, 20 µM). Non-stimulated pDCs were used as control. The cells were stained for IRF7. (**A**) Representative images of pDC phospho-IRF7 (green) and NUR77 expression (red); nuclei are counterstained by Hoechst 33342 (blue). (**B**,**C**) Quantification of the corrected total cell fluorescence (CTCF) for p-IRF7 and NUR77 staining revealed a marked inhibitory effect of oxLDL on CpG-mediated Ser471/2 phosphorylation of IRF7; this effect could be mimicked by Nur77 agonist LCA and rescued by co-treatment with the Nur77 ligand-binding domain antagonist TMPA. *** *p* < 0.001.

## Data Availability

All but the microarray data are presented in the manuscript and the Appendix A.

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
