# Peer review of "Low Density Lipoprotein Exposure of Plasmacytoid Dendritic Cells Blunts Toll-like Receptor 7/9 Signaling via NUR77"

_biomedicines, 2022, doi:10.3390/biomedicines10051152_

Round 1
Reviewer 1 Report
The authors studied the effect of the atherogenic lipoprotein, oxidized low-density lipoproteins (ox LDL) on the maturation of plasmacytoid dendritic cells (pDCs). The authors found that exposure of ox LDL and LDL led to pDCs lipid accumulation and down-regulation of the maturation markers. The authors concluded that a hyperlipidemic environment inhibits the maturation of pDCs and abrogates pDC-elicited T cell responses.
The weakness of this study is that the pathophysiological implication is unclear. The authors need to explain why ox LDL, generally recognized atherogenic lipoproteins, and “native” LDL, one of the physiological lipoproteins, led to pDCs being immature. How to reproduce hyperlipidemia in vitro experiments is also a critical issue.
Please modify minor issues:
Lines 79-80, (LPS; supplier)
Line 80, clone G.28.5 (agonist,
Figure 2, Y-axis label
Author Response
Reviewer 1:
General: this reviewer is thanked for his/her valuable remarks and suggestions. Please find below a point by point rebuttal of the comments
- The weakness of this study is that the pathophysiological implication is unclear. The authors need to explain why oxLDL, generally recognized atherogenic lipoproteins, and “native” LDL, one of the physiological lipoproteins, led to pDCs being immature. How to reproduce hyperlipidemia in vitro experiments is also a critical issue.
Based on an article published by our group in 2011 (Daissormont et al., Circ.Res. 2011; 109:1387-1395), we decided to investigate the effect of different atherogenic lipoproteins (LDL, oxLDL) on the functionality of circulating human pDC in this study: in our murine model we have previously shown that pDC exert their function primarily in the periphery by dampening CD4+ T-cell responses, thus functioning in an athero-protective manner (Daissormont et al., Circ.Res. 2011; 109:1387-1395). Overall, pDC depletion led to reduced CD4+ T cell priming in familiar hypercholesterolemia (FH)-similar conditions, which was ascribed to a lack of pDC-IDO activity.
This is in line with what we now observe for human pDC. Our current study points towards an immunosuppressive activity of human pDC under ex vivo hyperlipidemic conditions.
However, we are aware of atherogenic and T-cell mitogenic effects of pDC reported by different groups (Mafia et al., Arterioscler Thromb Vasc Biol. 2012;32:2569-2579; Döring Y et al., Circulation 2012; 125:1673-1683) in ApoE-knockout mice. Amongst others, effects were mediated by exposure of pDC to autoantigenic protein−DNA complexes, thus promoting atherosclerosis. Of note, concentrations used in the latter paper were above physiological range.
Under atherogenic conditions circulating levels of the lipoproteins LDL and oxLDL are increased. These can have an influence on the functionality of circulating immune cells, especially of the myeloid origin (unpublished data by our group). This finding and our previously published data indicating that “plaque-resident” pDC do not play a relevant role in inflammatory processes (Daissormont et al., Circ.Res. 2011; 109:1387-1395) encouraged us to investigate the specific effect of LDL/ oxLDL on circulating human pDC (ex vivo reflection of hypercholesterolemic conditions).
Indeed, we and others have shown in mouse and human, that pDC are not the prime source of IFN-α in plaque and plasma. Apparently, circulating IFN-α may be derived from macrophages (Goossens et al., Cell Metabolism 2010, 12(2):142-153; Kim et al., Circ.Res. 2018, 123(10):1127-1142). Moreover, we failed to demonstrate progressively increased expression of IFN-α (by microarray or real-time PCR analysis) by circulating pDC from atherosclerotic mice and by human pDC from patients with stable versus unstable disease and by unstable versus stable endarterectomy lesions, confirming that in chronic inflammatory processes such as atherosclerosis TLR7/9 activation of pDC is not very prominent (Daissormont et al., Circ.Res. 2011; 109:1387-1395). To demonstrate this, we specifically depleted pDC by the 120G8 mAb, which recognizes PDCA-1, a marker specifically expressed on mouse pDC: pDC activity and TLR 7/9 function appeared to be intact under conditions of hyperlipidemia. Though, baseline plasma IFN-α levels remained unchanged after pDC depletion. Only at a functional level, 120G8 treatment almost abrogated CpG-induced pDC activation in vivo, as judged by a significantly attenuated induction in plasma IFN-α release on CpG injection in 120G8 treated versus control mice. Overall, this suggests that an atherogenic stimulus per se does not increase IFN-α release by peripheral pDC and/or that pDC are under these conditions not the major source of circulating IFN-α. We added a sentence in the discussion (p.26, line 5-13) and in the introduction (p.4, final sentence first paragraph).
It would be of utmost interest in future experiments to investigate the influence of other plasma lipids (cholesterol, triglycerides) on the functionality of circulating human pDC. However, our study pointed to a role of NUR77 in LDL mediated suppression. Although cholesterol itself has not been reported to have high affinity for this nuclear receptor, we indeed cannot exclude that cholesterol metabolites produced in lipid laden pDC, rather than oxysterols entrapped in LDL, account for NUR77 activation. Though interesting, we felt that this goes beyond the key message of our study and will be subject for further study. A note on this is added in the discussion (page 28, final paragraph).
- Please modify minor issues:
- Lines 79-80, (LPS; supplier):
This information is now added in the revised manuscript (page 6).
- Line 80, clone G.28.5 (agonist):
The supplier info is now added in the revised manuscript (page 7).
- Figure 2, Y-axis label:
Figure 2 is now adjusted in the revised manuscript.
Reviewer 2 Report
The study by Anette Christ and colleagues is a well-performed analysis of how hyperlipidemia impacts human pDC activation, cytokine response and capacity to prime CD4+ T cells. The manuscript is well written and direct, and although the findings are not very novel, this study might be of interest for researchers in atherosclerosis and immunology field. It was previously described that the LXR pathway interferes with TLR7-induced NF-κB activation at different levels, including a transcriptional repression of p65 NF-κB subunit and a reduced phosphorylation of this NF-κB subunit (10.1182/blood-2016-06-724807).
Some comments:
- Is there any metabolic alteration in pDCs treated with Ox-LDL (either in glycolysis or FAO) that could complementary explain why this treatment impairs pDC maturation? (https://www.ncbi.nlm.nih.gov/pmc/articles/PMC5437952/).
-Line 43: the authors comment “we and others have demonstrated a role for pDCs in chronic inflammatory diseases such as atherosclerosis”. However, the results obtained by this group are contradictory to the findings of other groups. They should discuss this issue more deeply, since most groups have find that pDCs are detrimental to atherosclerosis ( 10.3389/fphys.2012.00230; https://doi.org/10.3389/fphys.2014.00279; https://rupress.org/jem/article/217/1/e20190459/132613/Type-I-interferons-in-atherosclerosisType-I ). It seems evident that type I IFN potentiates cardiovascular pathology, including atherosclerosis. Then, how the do the authors interpret their results on impaired IFNa secretion by pDCs treated with Ox-LDL?
-Figure 1. The authors may comment on the differences, if any, between Ox-LDL and LDL, and why have they decided to use Ox-LDL along the manuscript. They may also discuss or demonstrate that these effects are reproduced with other lipids (palmitate, Cholesterol–methyl-β-cyclodextrin).
-Figure 2B. Are the differences shown in this figure significant?
-Figure 4. Are the differences shown in this figure significant?
-Supplementary Figure 1D. Please, show unstimulated cells, and quantify co-localization.
Author Response
Reviewer 2:
General: this reviewer is thanked for his/her valuable remarks and suggestions. Please find below a point-by-point rebuttal of the comments
- Is there any metabolic alteration in pDCs treated with Ox-LDL (either in glycolysis or FAO) that could complementary explain why this treatment impairs pDC maturation? (https://www.ncbi.nlm.nih.gov/pmc/articles/PMC5437952/).
Inspired by this question, we have analyzed the expression of transcriptionally regulated rate limiting genes in pDC, using the Met-Flow panel (PMID: 32533056). As shown in the below figure, TLR9 activation (CpG) led to downregulation of phosphate metabolism (SLC20A1) and glucose uptake (SLC2A1=GLUT1), respectively the main glucose uptake transporters in pDC with a minor increase of ASS1 (arginosuccinate synthesis). No effects were seen on fatty acid synthesis (ACACA), fatty acid oxidation (CPT1A), oxidative phosphorylation (ATP5F1), oxidative stress (PRDX2), glycolysis (HK1, G6P) and PPP (G6P). This suggests that metabolic regulation in human pDC may differ from that in mouse pDC, as recently reported for macrophages (Verberk et al, Cell Reports, 2022, https://doi.org/10.1016/j.crmeth.2022.100192). While oxLDL exposure did not affect synthesis, it did further decrease oxPHOS activity (ATP5F1) and tended to increase IDH2 (Kreb’s cycle) in CpG stimulated cells. This information is now added to the results (Suppl fig. 3C and page 21).
- Line 43: the authors comment “we and others have demonstrated a role for pDCs in chronic inflammatory diseases such as atherosclerosis”. However, the results obtained by this group are contradictory to the findings of other groups. They should discuss this issue more deeply, since most groups have find that pDCs are detrimental to atherosclerosis ( 10.3389/fphys.2012.00230; https://doi.org/10.3389/fphys.2014.00279; https://rupress.org/jem/article/217/1/e20190459/132613/Type-I-interferons-in-atherosclerosisType-I ). It seems evident that type I IFN potentiates cardiovascular pathology, including atherosclerosis. Then, how do the authors interpret their results on impaired IFN-a secretion by pDCs treated with Ox-LDL?
We agree with the reviewer that our findings were pointing to a tolerogenic role of pDC (Ab depletion, LDLR-/-), whereas those of Doering et al (Circulation, 2012, 125:1673–1683) and Sage et al. (Circulation, 2014, 130:1363–1373) were pointing to a pro-inflammatory effect mediated by Cramp/DNA complex activation and MHCII antigen presentation, respectively. This discrepancy may reflect differences in the different mouse models (LDLR-/- vs ApoE-/- vs BMT), pDC depletion strategy (genetic vs antibody-dependent depletion) and disease stage as discussed earlier (Maffia et al., doi: 10.3389/fimmu.2017.00140).
On the second point, regarding the pro-atherogenic effects of type-I IFN: it is important to realize that macrophages are probably the major source of type-I IFN in plaque (Goossens et al, Cell Metab. 2010;12(2):142-1532010); single cell data have confirmed this notion and even showed that foamy macrophages produce less IFN-a than non-foamy macrophages (Kim et al. Circulation Research. 2018; 123:1127–1142), in line with our findings on lipid laden pDC. It is important to note that pDC are very scarce in plaque (Daissormont et al., Circ.Res. 2011; 109:1387-1395), and mainly reside in the adventitia, where they are not exposed to high oxLDL deposits.
Our findings may therefore be mainly relevant for pDC responses in circulation and lymphoid organs, rich in pDC. Indeed, we failed to demonstrate progressively increased expression of IFN-α (by microarray or real-time PCR analysis) by circulating pDC from atherosclerotic mice and by human pDC from patients with stable versus unstable disease and by unstable versus stable endarterectomy lesions, confirming that in chronic inflammatory processes such as atherosclerosis TLR7/9 activation of pDC is not very prominent (Daissormont et al., Circ.Res. 2011; 109:1387-1395). We have briefly elaborated on this in the discussion section (page 26).
- Figure 1. The authors may comment on the differences, if any, between Ox-LDL and LDL, and why have they decided to use oxLDL along the manuscript. They may also discuss or demonstrate that these effects are reproduced with other lipids (palmitate, Cholesterol–methyl-β-cyclodextrin).
We thank the reviewer for this important comment that we already addressed above:
Under atherogenic conditions circulating levels of the lipoproteins LDL and oxLDL are increased. These can have an influence on the functionality of circulating immune cells, especially of the myeloid origin (unpublished data by our group). This finding and our previously published data indicating that “plaque-resident” pDC do not play a relevant role in inflammatory processes (Daissormont et al., Circ.Res. 2011; 109:1387-1395) encouraged us to investigate the specific effect of LDL/ oxLDL on circulating human pDC (ex vivo reflection of hypercholesterolemic conditions). We agree with the reviewer that it would be of utmost interest in future experiments to investigate the influence of other plasma lipids (cholesterol, triglycerides) on the functionality of circulating human pDC. However, our study pointed to a role of NUR77 in LDL mediated suppression. Although cholesterol itself has not been reported to have high affinity for this nuclear receptor, we indeed cannot exclude that cholesterol metabolites produced in lipid laden pDC, rather than oxysterols entrapped in LDL, account for NUR77 activation. Though interesting, we felt that this goes beyond the key message of our study and will be subject for further study. A note on this is added in the discussion (p. 28, line 3-9).
- Figure 2B. Are the differences shown in this figure significant?
We have now added significance marks in the revised manuscript.
- Figure 4. Are the differences shown in this figure significant?
We have now added significance marks in the revised manuscript.
- Supplementary Figure 1D. Please, show unstimulated cells, and quantify co-localization.
This control experiment solely served for qualitative exploration of the lack of oxLDL uptake on CpG uptake and vice versa. As too few cells were contained per image (10-20) quantitative assessment of the level of colocalization is not meaningful. The qualitative character of this figure is now stressed in the results section (page 16).
Round 2
Reviewer 1 Report
I have no further comments.
Reviewer 2 Report
The authors have addressed most of my concerns with the original manuscript. I consider that the revised manuscript is ready for publication.